# Shedding Light on the Interaction of Human Anti-Apoptotic Bcl-2 Protein with Ligands through Biophysical and in Silico Studies

**DOI:** 10.3390/ijms20040860

**Published:** 2019-02-16

**Authors:** Joao Ramos, Jayaraman Muthukumaran, Filipe Freire, João Paquete-Ferreira, Ana Rita Otrelo-Cardoso, Dmitri Svergun, Alejandro Panjkovich, Teresa Santos-Silva

**Affiliations:** 1UCIBIO-NOVA, Departamento de Química, Faculdade de Ciências e Tecnologia, Universidade NOVA de Lisboa, 2829-516 Caparica, Portugal; jc.ramos@campus.fct.unl.pt (J.R.); muthu@fct.unl.pt (J.M.); f.freire@campus.fct.unl.pt (F.F.); jcp.ferreira@campus.fct.unl.pt (J.P.-F.); a.cardoso@campus.fct.unl.pt (A.R.O.-C.); 2European Molecular Biology Laboratory (EMBL), Hamburg Outstation, c/o DESY, 22067 Hamburg, Germany; svergun@embl-hamburg.de (D.S.); alejandro.panjkovich@embl-hamburg.de (A.P.)

**Keywords:** cell apoptosis, Bcl-2, venetoclax, BH3 mimetics, bioinformatics, in silico, biophysical methods, protein-ligand interactions, high throughput virtual screening

## Abstract

Bcl-2 protein is involved in cell apoptosis and is considered an interesting target for anti-cancer therapy. The present study aims to understand the stability and conformational changes of Bcl-2 upon interaction with the inhibitor venetoclax, and to explore other drug-target regions. We combined biophysical and in silico approaches to understand the mechanism of ligand binding to Bcl-2. Thermal shift assay (TSA) and urea electrophoresis showed a significant increase in protein stability upon venetoclax incubation, which is corroborated by molecular docking and molecular dynamics simulations. An 18 °C shift in Bcl-2 melting temperature was observed in the TSA, corresponding to a binding affinity multiple times higher than that of any other reported Bcl-2 inhibitor. This protein-ligand interaction does not implicate alternations in protein conformation, as suggested by SAXS. Additionally, bioinformatics approaches were used to identify deleterious non-synonymous single nucleotide polymorphisms (nsSNPs) of Bcl-2 and their impact on venetoclax binding, suggesting that venetoclax interaction is generally favored against these deleterious nsSNPs. Apart from the BH3 binding groove of Bcl-2, the flexible loop domain (FLD) also plays an important role in regulating the apoptotic process. High-throughput virtual screening (HTVS) identified 5 putative FLD inhibitors from the Zinc database, showing nanomolar affinity toward the FLD of Bcl-2.

## 1. Introduction

Cancer is one of the leading causes of death in humans, owing to the plethora of events responsible for its activation, the lateness of diagnosis, and the ineffectiveness of treatments available. In 2000, the hallmarks of cancer were postulated: sustaining proliferative signalling; evading growth suppressors; resisting cell death; enabling replicative immortality; inducing angiogenesis; activating invasion; and metastasis [1]. More recently, in 2011, Hanahan and Weinberg [2] extended this list by including evasion to immune destruction, and deregulating cellular energetics.

Tissue homeostasis comprises a strict balance between cell proliferation and cell death, and changes in the rate of the latter may implicate tumor formation [3]. As established previously, evasion or resistance to cell death is a hallmark of cancer, and this happens due to genetic mutations that alter either the expression or function of proteins [4]. Apoptosis, or programmed cell death (PCD), is a genetically defined mechanism that allows abnormal cells to commit suicide. It is a fundamental feature for multi-cellular organism survival, since it promotes the eradication of damaged cells which may interfere with the organism’s normal functioning and even promote tumor formation [5]. The apoptotic processes can be divided into two well-studied pathways, namely, intrinsic (or mitochondrial mediated) and extrinsic (or death receptor mediated).

One of the most well-known forms of the apoptotic intrinsic pathway takes place in the mitochondria and is governed by the stress-mediated release of cytochrome c to the cytosol that culminates in the formation of the apoptosome. The apoptosome activates initiator caspase 9, which leads to the activation of executioner caspase 3. The active caspase 3 is responsible for the cleavage of death substrates that lead to well-known processes of an apoptotic cell, including DNA fragmentation, nuclear fragmentation and membrane blebbing. [5]

B-cell lymphoma 2 (Bcl-2) family proteins are responsible for the homeostasis that dictates cell survival or cell death through the intrinsic pathway. The first member of this family to be found was the Bcl-2 protein, which was isolated from a t(14;18) chromosomal translocation derived from a human follicular lymphoma by Fukuhara and Rowley [6]. The remaining Bcl-2 family members were added by sequence homology identification, which enabled the postulation of four different domains, named Bcl-2 homology domains (BH1, BH2, BH3, BH4) [7,8]. The Bcl-2 family proteins can be divided into two main groups, according to their function: anti-apoptotic and pro-apoptotic. The anti-apoptotic members are Bcl-2, Bcl-w, Bcl-xL, Mcl-1 and A1, which are multi-BH domain proteins, including the four BH domains and a transmembrane domain (TM). The pro-apoptotic members comprise multi-BH domain effector proteins (Bax and Bak) and BH3-only proteins (Bad, Bim, Bid, Bik, Bmf, Hrk, Noxa and Puma) [9,10].

As the cell senses a death signal, BH3-only proteins, called sensitizers, sequester the anti-apoptotic proteins, such as Bcl-2. Unable to bind the BH3 domain of its pro-apoptotic partners, Bcl-2 is not capable of abolishing the apoptotic pathway. The pro-apoptotic proteins Bax/Bak form homo-oligomers in the mitochondria outer membrane (MOM), promoting its permeabilization to apoptogenic proteins, such as cytochrome c and second mitochondria-derived activator of caspases (SMAC) [9]. This phenomenon of cytochrome c displacement from the mitochondria to the cytosol leads to the apoptotic events described before. One may consider this mechanism as a population-equilibrium, since the higher availability of pro-apoptotic members causes the progression of apoptosis, while an excess of anti-apoptotic members avoids the apoptotic pathway, through the inhibition of their pro-apoptotic partners.

Bcl-2 protein is estimated to be overexpressed in almost as many as half of all human cancers [11]. Chromosomal translocation as a mechanism of Bcl-2 gene activation is associated with non-Hodgkin’s lymphomas, while the loss of endogenous miRNA and gene hypomethylation were reported in chronic lymphocytic leukemia (CLL) [12,13]. Moreover, Bcl-2 overexpression is correlated not only with non-Hodgkin’s lymphomas and CLL, but also small cell lung and breast cancers [14,15].

The role of Bcl-2 in the mentioned pathologies entails the dysregulation of the pro-apoptotic and anti-apoptotic equilibrium present in healthy cells. In cancer, this equilibrium is shifted toward the anti-apoptotic members that, in excess, inhibit the pro-apoptotic proteins. When faced with a death signal, the tumor cell cannot activate its apoptotic mechanism, which results in cell survival, proliferation and extensive therapeutic problems, since these cells become resistant to chemotherapeutics [16,17,18].

The human Bcl-2 protein has two reported isoforms, alpha and beta, with structures solved by nuclear magnetic resonance (NMR) by Petros et al. in 2001 [19]. The most common alpha isoform has 239 amino acids and comprises the four BH regions, a flexible loop domain (FLD) and the TM domain in a helical bundle fold. In this arrangement, the BH1-BH3 domains form a hydrophobic groove (binding groove or BH3-groove), while BH4 stabilizes the overall structure. The binding groove plays a crucial role in the anti-apoptotic activity of bcl-2, as it accommodates the BH3 domain of the pro-apoptotic partners (Bax/Bak) [20]. Therefore, while Bcl-2 poses as a potential pharmacological target for inhibition, BH3 mimetics are the main category of promising therapeutic agents explored so far [8,21,22,23].

The main representative of the BH3 mimetic drug molecules, which displayed significant therapeutic activity in clinical trials, is the arylsulfonamide-based compound, venetoclax, also known as ABT-199 [23,24,25,26]. In 2016, the EMA (European Medicines Agency) and the FDA (United States Food and Drug Administration) approved venetoclax for the treatment of patients with CLL.

Apart from the BH3 binding groove, another attractive drug-target region of Bcl-2 protein is the flexible loop domain (FLD), also called the regulatory domain. This ca. 100-residue loop is usually highly disordered and comprises a negative regulatory region, with residues 32 to 68, and a positive regulatory region, with residues 69 to 87 [27,28]. It plays a key role in regulating the apoptotic process through the interactions with several other proteins, namely c-Jun N-terminal kinase-1 (JNK-1), protein kinase C (PKC), caspase 3, mitogen-activated protein kinase (MAP kinase), FK506-binding protein 8 (FKBP38) [29]. This region contains phosphorylation sites in T56, T69, S70, T74 and S87, which model the activity of Bcl-2 [30,31]. DNA damage induced by the p53:Bcl-2 interaction was shown to be associated with the weaker Bcl-2:Bax interaction and enhanced PCD in a mechanism regulated by the FLD. The p53 protein can disrupt Bcl-2:Bax interaction by directly binding to the negative regulatory region of FLD [28]. To our knowledge, Taxol is the only drug reported, so far, that binds to the FLD, promoting the apoptotic process [32].

In the present work, we explore the human Bcl-2 protein-ligand interactions using biophysical and computational methods; experimental assays were carried out using a chimeric form of Bcl-2 where the FLD was replaced by a derivative loop from Bcl-xL to overcome protein precipitation, as described by Petros et al., while computational studies were carried out using chimeric and physiological forms, for comparison. We report evidence of great stability and conformational changes in Bcl-2 upon venetoclax binding, while the in silico results predict venetoclax binding at the atomic level and its impact on protein dynamics. Furthermore, we used HTVS-Docking analysis of physiological Bcl-2 form against the Zinc database [33] to screen for promising anti-cancerous molecules that target the FLD. Further in silico studies are also presented for non-synonymous single-nucleotide polymorphisms (nsSNPs) of Bcl-2 regarding stability and dynamical fluctuations in molecular dynamics (MD) simulations.

## 2. Results

### 2.1. Venetoclax-Bcl-2 Binding Using Biophysial Techniques

Thermal shift assay (TSA) is a fluorescence-based technique, routinely used in protein purification and crystallization, to assess protein stability in different conditions, namely in the presence of ligands, by following the protein thermal denaturation process [34,35,36,37,38]. Protein-ligand molar ratios of 1:2, 1:5 and 1:10 were tested using chimeric Bcl-2 to study the effect of venetoclax in the stability of the protein. In native conditions (i.e., in its final purification buffer) and in the presence of 2% DMSO, our Bcl-2 chimeric construct shows a melting temperature (T_m_) of approximately 65 °C. Meanwhile, upon incubation with venetoclax, this value increases to 83 °C. A significant shift in the protein’s T_m_ of 18 °C is observed in the presence of the ligand (Figure 1). This drastic change in protein resistance to thermal denaturation indicates a strong stabilizing interaction between venetoclax and Bcl-2.

Urea polyacrylamide gel electrophoresis (Urea PAGE) allows the detection of different conformational states of a protein upon interaction with a co-factor or ligand [39,40,41,42]. This electrophoresis is carried out in the presence of 6 M urea, which is a concentration that promotes protein denaturation. However, upon ligand binding, a protein may alter to a closed conformation, which enables larger electrophoretic mobilities. For the chimeric Bcl-2 incubated with venetoclax, a higher electrophoretic mobility is observed when compared with the ligand free form (Figure 1). This behavior suggests that the interaction of venetoclax with Bcl-2 enhances significantly its stability and resistance to urea denaturation, and putatively promotes a conformational change of the protein.

To elucidate the possible conformational changes of Bcl-2 upon venetoclax binding, Small angle X-ray scattering (SAXS) data was collected in the presence and absence of the ligand for the chimeric form. The scattering data of the free-form indicate a monomeric globular protein with some degree of aggregation that is not present in the ligand-bound form. Size exclusion chromatography combined with SAXS (SEC-SAXS) was used to collect scattering data of the protein while removing aggregates. The data collected for the ligand-free and ligand-bound forms are very similar, suggesting that the ligand dissociated from the protein during the elution experiment or the conformational changes are not so pronounced to detect by SAXS. The data suggest that the Bcl-2:venetoclax complex has a radius of gyration (Rg) of 1.8 nm and maximum particle size (Dmax) of ~5.8 nm. This Rg value corroborates the average Rg value obtained from MD simulation of physiological Bcl-2 form and chimeric Bcl-2 form. Although the putative conformational change could not be inferred, we see a very good fit between the Bcl-2:venetoclax structure (obtained by molecular docking, as described later) and the corresponding SAXS data, suggesting that the presence of the ligand stabilizes the protein and decreases the interparticle interactions. SAXS data was in agreement with MD simulation of venetoclax-bounded chimeric as well as physiological Bcl-2 forms. Together the results of SAXS and MD simulation suggest that the ligand enhances the structural stability and integrity (Appendix A) of Bcl-2 protein.

### 2.2. Molecular Docking of the Bcl2:Venetoclax Complex

To assess the binding mode of venetoclax toward Bcl-2, molecular docking calculations (Figure 2 and Figure 3) were also performed, using both its chimeric and physiological forms. Venetoclax binds strongly to the Bcl-2 binding groove with a predicted binding free energy (ΔG) of −10.24 kcal/mol for the physiological Bcl-2 and −11.35 kcal/mol for the chimeric form. Both docking calculations revealed that venetoclax occupies the BH3 binding pocket with common interacting residues F104, R107, Y108, and G145. According to the docking results with chimeric Bcl-2, venetoclax interacts with F112, T132 and E136, which are not included in the binding network established by navitoclax in the RCSB-PDB crystal structure 4LVT [27]. Moreover, the Q99 residue of physiological Bcl-2 form makes two hydrogen bonds with N15 and O40 atoms of venetoclax with the bond distance of 3.14 Å and 2.9 Å, respectively. Along with this, a few new hydrophobic interacting residues were also observed in physiological Bcl-2 form, namely L95, R98, L201, G203 and P204. The high number of interaction sites suggests a tight binding between physiological Bcl-2 and venetoclax. 

### 2.3. Molecular Dynamics

#### 2.3.1. Physiological and Chimeric Bcl-2: Venetoclax Complexes

Molecular dynamics presents an opportunity to study protein dynamics in a time- and cost-efficient way, adding to the knowledge retrieved from docking calculations, where the target is usually considered as a rigid identity. In this way, MD simulation is an interesting computational tool to understand the stability, flexibility, folding, and the dynamic behavior of proteins and various bio-molecular complexes (protein-protein, protein-small molecule, protein-carbohydrate and protein-lipid) at the same time with a regular time scale. Several essential structural parameters are obtained from this analysis, such as root mean square deviation (RMSD), root mean square fluctuation (RMSF), radius of gyration (Rg), intra-molecular hydrogen bonds, inter-molar hydrogen bonds, solvent accessible surface area (SASA), secondary structure and residue occupancy probability. Bcl-2:venetoclax complexes—chimeric and physiological forms—attained their stabilized state at 10 ns, maintaining it until the end of the MD simulation. The average RMSD value was from 0.45 to 0.5 nm indicating that the complexes were stable and had converged properly. During the MD simulations, we could observe that the physiological form of Bcl-2 showed higher RMSD values than the chimeric form, likely due to the instability conferred by the large flexible loop. Nonetheless, after reaching 25 ns, both systems were equilibrated, having similar RMSD values, showing that the binding of the drug molecule is similar in both cases (Appendix A). Moreover, in comparison with the ligand-free Bcl-2, the ligand-bound physiological form is highly stable due to the consistent RMSD values from 15 ns of the MD simulation onwards (Appendix A). Regarding the Rg analysis, after 10 ns, a consistent value of 1.6 nm is observed in the chimeric Bcl-2:venetoclax complex, which is in agreement with the SAXS data. Meanwhile, due to the longer flexible loop present in the physiological Bcl-2 form, higher average Rg values of around 1.8 nm are evident. Despite this, the overall Rg results show that the protein is properly folded and in regular geometry during the interaction with the inhibitor. Parameters such as the Rg indicate that the presence of the compounds enhances protein compactness and folding properties (Appendix A). The formation/deformation of intermolecular hydrogen bonds is also an essential property in protein-ligand complexes’ MD simulations. An average of 2 to 3 hydrogen bonds are observed between chimeric Bcl-2 protein and venetoclax, in a regular interval denoting that the ligand interacted and stayed bounded in the binding groove of Bcl-2. As described in the previous section, a higher number of intermolecular hydrogen bonds (average of 5 to 7) were observed between the physiological Bcl-2 form and the inhibitor (Appendix A). The SASA profiles of both complexes changed due to inhibitor binding. The 30 ns MD simulation performed for the protein-ligand complexes showed a preferential ligand interaction with residues in the Bcl-2 binding groove, as expected. Ligand atoms interacted with these residues at distances shorter than 3 Å. The residue occupancy probability map showed that venetoclax interacts significantly with the BH3 binding pocket and, to a much lower extent, with the FLD of Bcl-2 protein (Appendix A). Essential dynamics analysis supported that the ligand-bound form had slightly higher flexibility for the physiological Bcl-2 compared to the chimeric form due to the presence of a long and flexible loop (Appendix A). However, the energy difference between the ligand free and the bound states is due to the venetoclax interaction, which indicates an alteration in protein stability, and potentially conformation as well. When compared with the ligand free chimeric and physiological Bcl-2 forms, there is a stabilization in the presence of venetoclax, of −570630 kJ/mol and −652857kJ/mol, respectively. These results are explained by the large aromatic chain of venetoclax which interacts strongly with the numerous hydrophobic residues in the Bcl-2 groove, as suggested by the molecular docking calculations. The presence of the ligand influenced significantly the protein’s energy potential and flexibility.

#### 2.3.2. Non-Synonymous Single-Nucleotide Polymorphisms in Physiological Bcl-2

Concordance analysis using 8 different in silico programs (listed in the Materials and Methods section) was used to understand the functional consequences and putative phenotypic effects of nsSNPs, which were classified as deleterious or non-deleterious. This analysis yielded a list of 12 nsSNPs with a predicted deleterious phenotype: G8E, D34Y, A43T, H94P, L97P, F104S, S105F, S105P, R129C, G203S, R207W and G233S (details in Appendix A). Among the 12 putative deleterious nsSNPs here predicted, A43T was reported to be associated with esophageal cancer [43] and hepatocellular carcinoma (HCC) [44]. Some of the nsSNPs are also putative post-translational modification (PTM) sites for O-linked glycosylation and phosphorylation (Appendix A): F104S (rs751038951), D34Y (rs540701354) and G203S (rs148811059) can be phosphorylated; A43T (rs1800477), can be both phosphorylated and O-link glycosylated. There are many examples in the literature where gain or loss of post-translational modification affects protein functions/regulation, and is considered to be a hallmark of pathogenesis [45,46,47,48,49,50,51]. Our analysis suggests that these nsSNPs are likely to hinder the protein physiological function.

The MD simulation analysis results (Appendix A) suggest that G8E, L97P, F104S, S105F structures are generally more flexible than the physiological form of Bcl-2, with variable interdomain distances. Moreover, the impact of the nsSNPs on venetoclax binding was also investigated, since a specific phenotype may interfere with the effectiveness of a standard treatment, especially since some of the deleterious nsSNPs here suggested are involved in Bcl-2:venetoclax binding (F104 and G203). The binding free energies predicted using Auto Dock Vina [52] docking calculations (Table 1) showed that venetoclax binding is generally favored regardless of the presence of the mutations, probably due to the large number of interaction sites between Bcl-2 and this drug.

### 2.4. Virtual Screening for Physiological Bcl-2: Targeting The FLD

As mentioned previously, p53 binding to the FLD of Bcl-2 weakens the Bcl2:Bax interaction and results in the enhancement of the apoptotic pathway. The instability that this region promotes in Bcl-2 is a major limitation for experimental studies on recombinant Bcl-2. Therefore, the chimeric form reported by Petros et al., with enhanced stability is used in the present study. In silico studies offer the possibility to investigate the interaction of the FLD with small molecules, paving the way for finding new Bcl-2 inhibitors. These new promising inhibitor candidates can be tested later for in vivo activity and can be considered for combination therapy. Hence, we attempted to find candidates to target the physiological Bcl-2 form, by performing a virtual screening using the Zinc database [33]. We visualized 1000 possible docking solutions of various molecules in PyMOL [53] to locate the binding modes in the physiological Bcl-2 structure. Among the docking solutions, 68% of molecules were oriented towards the FLD, whereas the remaining 32% were located in the binding groove (formed by BH1, BH2 and BH3 domains) of Bcl-2. The docking solutions were submitted to the CDRUG webserver [54] to predict their putative anti-cancer properties. Among the 7 molecules that might have possible anti-cancer properties, 5 have a higher affinity for the FLD, and only 2 are oriented in the binding groove. The 5 hits that bind to the FLD are in the same orientation as Taxol, known to bind to the domain [32], suggesting that these molecules will also impair the FLD function (Figure 4 and Figure 5). 

Moreover, in comparison with Taxol, these five putative molecules show a stronger binding affinity (inhibition in the nM range) against FLD (Table 2): “ZINC09475116” was classified as a top first binder, showing estimated free energy of binding of –10.9 kcal/mol (predicted Ki: 10.2 nM). This molecule forms hydrophobic interaction with the T74 and S87 residue of FLD (Figure 5), which is an important residue for regulating anti-apoptotic function of Bcl-2 [30,31]. As Taxol, the following molecules ZINC22238492, ZINC20149102, ZINC04921974 also interact with T74 (Appendix A). These results suggest that these molecules are promising candidates to inhibit Bcl-2, targeting the FLD.

## 3. Discussion

In 2013, Souers et al. reported venetoclax as a potent and selective inhibitor for Bcl-2, which could establish stronger interactions with Bcl-2 when compared with Bcl-xL and Bcl-w, thus diminishing platelets’ death. To quantify the binding affinity of venetoclax toward Bcl-2, the authors used time-resolved fluorescence energy transfer competition binding assays. This technique couldn’t determine with precision the inhibitory constant associated, reporting a value < 0.01 nM. Since then, no other biophysical characterization has been reported for the sub-nanomolar binding toward Bcl-2 of this drug, which is already approved for treating patients in Europe and the United States.

Two significant properties are concomitant with ligand binding to proteins: stability and conformation. The interactions established in a protein-ligand complex generally result in global energy minimization, increasing protein stability, and might implicate conformational changes, due to ligand accommodation and displacement of water molecules.

The TSA results show a remarkable shift in Bcl-2 T_m_ upon interaction with venetoclax of 18 °C. This increase in resistance to thermal denaturation suggests that venetoclax binds strongly to Bcl-2, promoting the maintenance of the protein folding. When compared with the TSA results reported for a nanomolar inhibitor of Bcl-2, Disarib (*K_d_* = 28 nM) [38], the ΔT_m_ of venetoclax is almost 4-fold. This observation corroborates the strong binding affinity reported by Souers et al. (*K_I_* < 0.01 nM).

Concomitant with the increase in protein stability, the interaction between venetoclax and Bcl-2 might implicate conformational changes in the protein tertiary structure. Urea PAGE and SAXS measurements were performed to assess this hypothesis. The urea electrophoresis revealed a significant increase in electrophoretic mobility of Bcl-2 upon incubation with venetoclax. This is in agreement with the strong binding reported for venetoclax and validated by the TSA, indicating that the protein assumes a more stable conformation upon venetoclax binding. However, since chemical denaturation is the methodology used, protein stability could be a more relevant factor in electrophoretic mobility than protein conformation. The electrophoretic results may suggest, as well, that the ligand free chimeric Bcl-2 form has poor stability and thus resistance to denaturation, while the ligand-bound Bcl-2 is more stable and may display a larger mobility in the gel. To shed light on the hypothesis that Bcl-2 undergoes significant conformational alterations upon binding venetoclax, SAXS data was collected on ligand free and ligand-bound samples. The results indicate similar folding for both free and venetoclax-bound states. Considering the strong interaction between Bcl-2 and venetoclax reported and validated by the TSA and the Urea PAGE, it seems unlikely that the ligand would dissociate from Bcl-2 upon elution in the SEC. Therefore, although venetoclax binding to Bcl-2 appears to increase drastically protein stability, the protein folding remains native-like without detectable conformational changes.

Since venetoclax was derived from the navitoclax (ABT-263) scaffold, it was expected to bind in the same Bcl-2 groove, establishing a few new interactions with other protein residues which dictate its selectivity when compared to Bcl-xL and Bcl-w. In agreement with the binding affinity reported by Souers et al. and the TSA and electrophoretic results here presented, highly favoured interactions of venetoclax toward chimeric and physiological Bcl-2 were predicted by molecular docking, of −11.35 kcal/mol and −10.24 kcal/mol, respectively. The docking calculations for the chimeric Bcl-2 suggest that venetoclax interacts with F112, T132 and E136 of Bcl-2, which do not belong to the binding network found for the Bcl-2:navitoclax complex (PDB code 4LVT). In fact, these residues are spatially close and seem to influence greatly the venetoclax binding mode through hydrophobic interactions, when compared to navitoclax. In the case of the physiological Bcl-2 form, the docking calculations also show interactions with L95, R98, Q99, L201, G203 and P204, in comparison with the docking of the chimeric form. The high number of interaction sites suggests a tight binding between physiological Bcl-2 and venetoclax.

The structural alignment of Bcl-2 with Bcl-xL (PDB [56] ID: 2LPC [57]) and Bcl-w (PDB [56] ID: 1MK3 [58]), (Appendix A) through the structure comparison tool provided in the PDB [56], showed that T132 is not conserved in these Bcl-2 homologues, which leads to the hypothesis that this residue is pivotal for the venetoclax specificity toward Bcl-2. Moreover, in Bcl-xL and Bcl-w, T132 is replaced by Q85 and Q80, respectively. In spite of having similar chemical properties, glutamine has a longer side-chain, which would clash with venetoclax binding mode toward Bcl-2, and thus result in a less favorable conformation for Bcl-xL and Bcl-w binding. This dictates a weaker binding affinity for Bcl-2 homologs, justifying the drug’s reported Bcl-2 selectivity of over three orders of magnitude.

Complementing the in silico analysis provided by the molecular docking, MD simulations showed a decrease in potential energy of the chimeric and physiological Bcl-2:venetoclax complexes. This increase in Bcl-2 stabilization further corroborates our experimental results and shows the impact of venetoclax binding on protein dynamics.

MD simulations of the physiological form of Bcl-2 protein with each nsSNP were performed in order to study the effect of these mutations in protein dynamics. These mutations may implicate differences in protein stability, structure and function, leading in some cases to diseases [59]. An identification of nsSNPs responsible for a specific pathogenic state with experimental techniques is a costly and time-consuming process. Concordance analysis using several in silico tools with sequence and structure-based approach is an alternate way to identify and differentiate the tolerant and intolerant SNPs [60]. The MD simulations revealed that structures with the nsSNP of G8E, L97P, F104S, S105F generally are more flexible than the physiological form of Bcl-2. Despite this, a study of the impact of the nsSNPs on the interaction of Bcl-2 with venetoclax is of great therapeutic value. The molecular docking calculations showed that the nsSNPs do not disturb the favorable and effective interaction of venetoclax toward Bcl-2. Thus, in patients with these mutations, venetoclax should remain to be considered as an effective drug for treatment. Our future work will explore the impact of these nsSNPs with other Bcl-2 inhibitors reported so far in order to see the effect of mutations on inhibitor binding.

Despite venetoclax being the most potent inhibitor available of Bcl-2, there are additional opportunities to target Bcl-2, namely through the interaction with its regulatory FLD. As mentioned previously, Taxol is the sole reported example of the use of this strategy [32]. Our virtual screening, using the physiological Bcl-2 modeled structure, yielded 5 promising candidates that also target the FLD, with nanomolar affinity constants predicted. These hits can thus be considered for a structure-based drug design pipeline, leading to in vitro and in vivo assays that will result in their characterization and optimization aiming at their use for therapeutic purposes.

## 4. Materials and Methods 

### 4.1. Chimeric Bcl-2 Expression and Purification

The studied Bcl-2 chimeric construct, purchased from NZYTech, Lda (Lisbon, Portugal), lacks loop 2 (residues 34–92) from physiological Bcl-2. Instead, the protein has a derivative loop of the homolog anti-apoptotic protein Bcl-xL (residues 35–50), as reported by Petros et al. The recombinant protein was cloned in a pET28a(+) vector (Novagen, Wisconsin, WI, USA) with a six His-tag in its *N*-terminus. Protein overexpression was induced with Isopropyl β-d-1-thiogalactopyranoside (IPTG) at a final concentration of 0.5 mM in *E. coli* BL21 (DE3) cells grown in LB medium. Ultrasounds were used for cell lysis and the soluble fraction was purified by Ni^2+^-affinity chromatography. The Ni-NTA column (HisTrap^TM^ HP, GE Healthcare Life Sciences, Pittsburgh, PA, USA) was equilibrated with 50 mM HEPES pH 9.0, 300 mM NaCl, 10 mM imidazole and 1 mM β-mercaptoethanol. The recombinant protein was eluted with a 0.5 M imidazole gradient. The six His-tag fragment was cleaved by incubation with 1 mg of bovine plasma thrombin (Sigma-Aldrich, Missouri, MO, USA) in an overnight dialysis, at 4 °C, against 50 mM HEPES pH 9.0, 300 mM NaCl and 1 mM DTT. Pure protein was obtained after a final size exclusion chromatography using a superdex-75 10/300 GL column (GE Healthcare Life Sciences, Pittsburgh, PA, USA), equilibrated in 50 mM HEPES pH 9.0, 500 mM NaCl and 1 mM DTT.

### 4.2. TSA

TSA was performed to assess protein stability upon venetoclax incubation in protein-ligand molar ratios of 1:2, 1:5, and 1:10 in 2% DMSO. Additionally, two controls were set up, with and without 2% DMSO in the protein buffer. The assays were completed in MicroAmp® fast 96-well reaction plates (Applied Biosystems^TM^, ThermoFisher Scientific, Waltham, MA, USA), using a total sample volume of 20 μL, containing 2 μL of protein at 10 μM, 10 μL of buffer (control) or ligand solution, 3 μL of protein thermal shift dye (ROX^TM^) and 5 μL of dye buffer solution. The TSA were performed in 2 min cycles of 1% increments between 25 °C and 95 °C in a StepOnePlus^TM^ Real-Time PCR System (Applied Biosystems^TM^, ThermoFisher Scientific, Waltham, MA, USA). Data processing and analysis were performed with Protein Thermal Shift^TM^ software (Applied Biosystems^TM^, ThermoFisher Scientific, Waltham, MA, USA).

### 4.3. Urea PAGE

Urea PAGE was performed using a Novex 6% Tris-TBE urea mini gel in a XCell SureLock^TM^ Mini-Cell from Invitrogen (ThermoFisher Scientific, Waltham, MA, USA). Running buffer was diluted 1:5 from 89 mM Tris base and 89 mM boric acid, while the sample buffer used contained 45 mM Tris base and 45 mM boric acid, 6% FicollR Type 400, 3.5 M urea and 0.005% bromophenol blue.

Bcl-2 was incubated overnight at 4 °C with venetoclax in a 1:5 protein-ligand molar ratio in 5% DMSO. The protein-ligand solution was concentrated by centrifugation in an Amicon® Ultra-15 Centrifugal Filter Unit with UltraCel-10 Membrane. The loaded samples had 5 μL of ligand free and ligand incubated protein at 15 mg/mL and 5 μL of sample buffer. The urea gel was subjected to 180 mV, 40 mA for 1 h.

### 4.4. SAXS

SAXS data was collected at 0.1241 nm, at 283 K, at EMBL P12 beamline, DESY, Hamburg, Germany, using chimeric Bcl-2 at a concentration between 10 and 0.1 mg/mL. Data was collected for the ligand-free form of the protein and after overnight incubation with venetoclax, as described in the urea PAGE methods subsection, above. To remove the excess of ligand, the sample was dialyzed prior to data collection.

High and low concentration curves were merged to counter concentration effects such as interparticle interference using the program PRIMUS from the ATSAS package [61]. GNOM [62] was used to obtain the P(r) and determine the corresponding and values. 

### 4.5. Molecular Docking

Docking calculations of venetoclax into the NMR structure of Bcl-2 (PDB ID: 1GJH, chimeric form of Bcl-2) were performed using the program AutoDock 4.2 [63]. Initially, the three-dimensional structure of Bcl-2 was subjected to protein preparation steps which include: (a) addition of polar hydrogens, (b) merging non-polar hydrogens, (c) assignment of Kollman charges, and (d) conversion into PDBQT format. Afterward, the three-dimensional structure of the inhibitor venetoclax was retrieved from PubChem [64] database (CID 49846579) followed by (1) addition of Gasteiger charges, (2) addition of polar hydrogens, (3) merging of non-polar hydrogens, (4) setting up rotatable bonds, and (5) saving into PDBQT format. Both protein and ligand preparation steps were carried out with Molecular Graphics Laboratory (MGL) Tools [65]. Once protein and ligand preparation was completed, the receptor grid map was generated based on the BH3 domain binding pocket using the AutoGrid program (X = 81 Å, Y = 88 Å, Z = 75 Å number of points, grid spacing of 0.375 Å and grid centered at X = 11.230 Å, Y = -5.910 Å, Z = 2.270 Å). In the AutoDock calculations, the Lamarckian Genetic Algorithm (LGA) was used with the parameters of 1000 docking trials, 150 population size, 2,500,000 maximum number of energy evaluations, 27,000 maximum number of generations, 0.02 mutation rate and 0.8 cross-over rate. To understand the inhibitory effect of venetoclax toward the physiological form of Bcl-2, the docking calculation was performed with a full-length model of the physiological Bcl-2 form (full-length model generated from the Phyre2 web server [66]). The docking calculations of physiological Bcl-2 form with venetoclax were performed using a similar procedure as for the chimeric form, varying the grid parameters (X = 81 Å, Y = 88 Å, Z = 75 Å number of points, grid spacing of 0.375 Å and grid centered at X = 1.230 Å, Y = −4.910 Å, Z = −1.730 Å). The best protein-ligand complex was selected based on the estimated free energy of binding (ΔG), estimated inhibition constant (Ki) and the highest number of populations (largest cluster). Both docking results were analyzed and compared using three different programs, namely MGL Tools [65], PyMOL Version 1.3 [67] and LigPlot+ Version 1.4.5 [68]. Moreover, the Drug discovery@TACC web portal (https://drugdiscovery.tacc.utexas.edu/#/home) was used to perform the high throughput virtual screening analysis of physiological Bcl-2 form against the Zinc database [33] (library of ~642,759 drug-like molecules) using Auto Dock Vina [52] to identify the putative FLD inhibitors. The steps involved for protein preparation was mentioned above. Once virtual screening results were obtained, they were analyzed by using CDRUG webserver [54] and PyMOL [53] program.

### 4.6. Molecular Dynamics

#### 4.6.1. Physiological And Chimeric Bcl-2:Venetoclax Complexes

To assess the impact of venetoclax binding toward the chimeric Bcl-2 (PDB code, 1GJH) as well as the physiological Bcl-2 form, in the present study, MD simulations were performed using the Gromacs software version 5.0.5 [69]. 

The full-length model of human Bcl-2 was predicted using the intensive mode protocol of Phyre2 [66]. The three-dimensional structure of physiological Bcl-2 form was modelled based on 3 different templates (PDB ID: 2XA0, 1G5M, 2O2F). As there is no template available for the FLD region of Bcl-2, this portion was modeled by ab initio methods. The ligand-free Bcl-2 MD simulation was performed with both the chimeric and the physiological forms, using the OPLS-AA/L all-atom force field [70] and the default cubic box parameters. In the case of the physiological Bcl-2 form, 15,049 explicit flexible Simple point charge (SPC) water molecules [71] were generated and one was replaced by a sodium ion to balance the global net charge of −1.00 e, whereas 12,744 water molecules were added initially and nine sodium ions were added to neutralize the system’s charge of the chimeric Bcl-2 form. After energy minimization and temperature and pressure equilibration of the systems, the MD simulations were carried out at physiological temperature (37 °C) and atmospheric pressure (1.0 bar). The MD simulations of the complexes Bcl-2:venetoclax (venetoclax bound to the chimeric and physiological Bcl-2 forms) were performed using the GROMOS96 54a7 force field [72,73], the venetoclax topology file generated by automated topology builder (ATB version 2.0 [74]) and the default cubic box was used. In the chimeric Bcl2:venetoclax simulation, 12,911 explicit flexible SPC water molecules were generated and nine were replaced by sodium ions to balance the global net charge of −9.00 e. For the physiological form Bcl2:venetoclax simulation, 14,990 water molecules were added to the system and it attained the neutralized state without further addition of counter ions. Afterward, energy minimization and temperature and pressure equilibration followed by MD simulations carried out for 30 ns, at physiological temperature and atmospheric pressure. The results were analyzed through various built-in functions from the Gromacs software [69].

#### 4.6.2. Non-Synonymous Single-Nucleotide Polymorphisms in Physiological Bcl-2

The nsSNPs of the Bcl-2 protein were retrieved from dbSNP (NCBI, *URL: https://www.ncbi.nlm.nih.gov/projects/SNP/*) database [75,76], filtering the search with the following criteria: *Homo sapiens* (Organism), SNP (Variation Class), missense (Function Class), by-1000 Genomes, by-cluster and by-frequency (Validation Status). This search resulted in 36 SNPs retrieved for isoform alpha (UniProtKB P10415, *https://www.uniprot.org/uniprot/P10415*) and 3 non-redundant additional SNPs for isoform beta (UniProtKB P10415-2, *https://www.uniprot.org/uniprot/P10415#P10415-2*). After gathering 39 different Bcl-2 nsSNPs, a concordance analysis was performed using 8 different programs (I-Mutant [77], Panther [78], SNP&Go [79], SIFT [80], Provean [81], Polyphen 2.0 [82], nsSNPAnalyzer [83] and PhD-SNP [84]), in order to assess the potential phenotypic effect (deleterious or non-deleterious) of these mutations. These programs attempt to predict the SNPs impact on protein stability and function, using the ProTherm [85] database (Thermodynamic database for Proteins and Mutants, *URL: http://www.abren.net/protherm*) and two computational methods, namely, machine learning and hidden Markov models (HMM). We classified the nsSNPs in to deleterious or non-deleterious by comparing results obtained from all the programs and when concordance was obtained for at least 5 out of 8 programs used. This analysis resulted in a group of 12 nsSNPs out of 39. Afterwards, the effect of the 12 nsSNPs on structure stability was predicted using 7 different in silico programs, such as I-Mutant [77] mCSM [86], SDM [87], DUET [88], MuPro [89], INPS-MD [90] and iStable [91]. Protein sequence analysis was also performed for these 12 nsSNPs along with the physiological form of Bcl-2 protein using Expasy Proteomics Suite (URL: *http://www.expasy.org/tools*) in order to understand the effects of the mutations on the protein physico-chemical properties and post-translational modification processes. To understand the impact of nsSNPs on Bcl-2 structure and stability, through MD simulation, it is necessary to have its full-length model. The steps for prediction of full-length model was explained in the previous section. As physiological Bcl-2 form, 80 ns MD simulation was performed for 12 nsSNPs of Bcl-2 variants and the comparison between the wild-type and the nsSNPs forms was achieved through the various built-in functions of Gromacs [69]. For that purpose, their PDB coordinates were generated through of mutation of the physiological form of Bcl-2, using Coot software version 0.8.2 [92]. The parameters used in the MD simulations specific for each nsSNPs, such as the number of SPC water molecules in the box, the system global net charge and the type and number of counter-ions added for electrostatic equilibration are presented in Appendix A. Furthermore, to understand the structural and functional effect of the 12 nsSNPs on the physiological Bcl-2 form binding to venetoclax, molecular docking calculations were performed using Auto Dock Vina [52] with the following parameters: Grid box size: X = 47.62 Å, Y = 50.25 Å, Z = 43.12 Å; Box Centred: X = 39.91 Å, Y= 38.43 Å and Z = 42.37 Å). Docking results were analyzed using 3 different programs namely Auto Dock Tools [65], PyMOL [53] and LigPlus [93].

## 5. Conclusions

The most potent commercially available inhibitor of Bcl-2, venetoclax (ABT-199), interacts strongly with Bcl-2, resulting in considerable variations in protein stability. Through biophysical and computational methods, this phenomenon was observed and helped to shed light in Bcl-2 dynamics upon venetoclax binding. The significant binding affinity of the small molecule was corroborated by molecular docking calculations (ΔG: −10.24 kcal/mol for physiological and −11.35 kcal/mol for chimeric forms of Bcl-2) and the interactions which yield the fundamental Bcl-2 specificity of venetoclax were also identified. The TSA showed a significant T_m_ shift of 18 °C, indicating an increase in protein stability upon ligand binding which found correspondence in the essential dynamics analysis of ligand-bound chimeric Bcl-2 form and in the SAXS data. Urea PAGE suggests that the venetoclax binding affects Bcl-2 resistance to urea denaturation, by maintaining the native protein folding, while the ligand-free form shows high susceptibility to unfolding. Moreover, the computational study revealed that the binding of venetoclax towards chimeric and physiological Bcl-2 form is strong, and independent of the presence of the FLD. On the other hand, the nsSNPs of Bcl-2 variants showed a limited impact in protein stability and folding, which dictated a non-significant variation in binding affinity of venetoclax predicted by molecular docking studies, compared to the wild-type Bcl-2. Further experimental validation is underway in order to understand the impact of the most relevant nsSNPs, identified by in silico studies, for inhibitor binding. Moreover, 5 promising molecules were identified from HTVS-docking analysis to target the FLD of the physiological form of Bcl-2. These could be the starting point to develop more potent, and safer anti-cancerous molecules to modulate Bcl-2′s protein-protein interaction networks through the FLD.

## Figures and Tables

**Figure 1 ijms-20-00860-f001:**
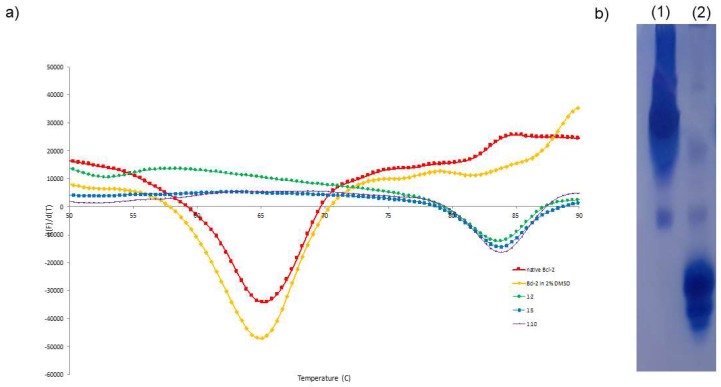
(**a**) TSA of chimeric Bcl-2 and incubated with venetoclax in different protein-ligand molar ratios: Bcl-2 in red, Bcl-2 with 2% DMSO in yellow, 1:2 in green, 1:5 in blue, 1:10 in pink color. Upon ligand incubation, the Bcl-2 T_m_ shifted significantly toward higher temperatures, representing a ΔT_m_ = 18 °C. (**b**) Urea polyacrylamide gel of (1) Ligand free form of chimeric Bcl-2 and (2) incubated with chimeric Bcl-2:venetoclax. A great difference in electrophoretic mobility is observed in the presence of ligand, indicating relevant modifications in protein folding.

**Figure 2 ijms-20-00860-f002:**
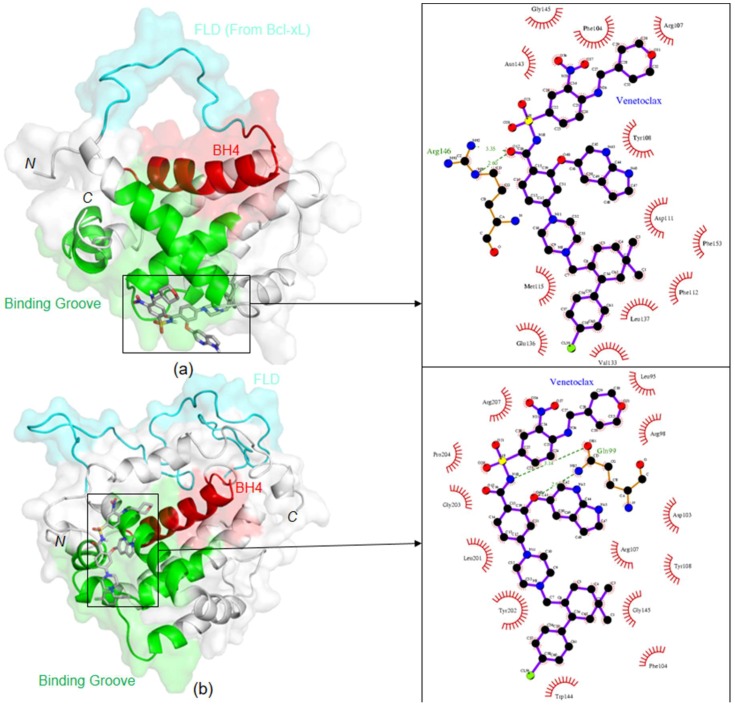
Docking orientation of venetoclax in (**a**) chimeric and (**b**) physiological Bcl-2 forms, and schematic representation of corresponding protein-ligand interactions, on the right. Color codes: FLD in cyan; binding groove formed by BH1, BH2 and BH3 domains in green; BH4 domain in red.

**Figure 3 ijms-20-00860-f003:**
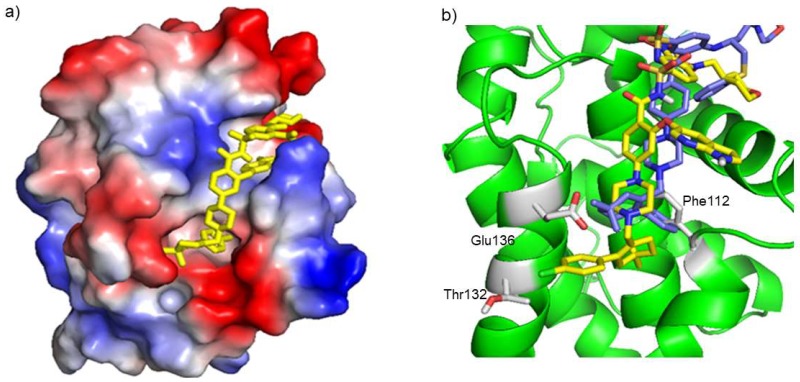
(**a**) Representation of venetoclax binding mode toward chimeric Bcl-2 predicted by AutoDock. Bcl-2 electrostatic surface potential is represented, where negative charges are shown in red and positive charges in blue. Venetoclax is represented by sticks in yellow. (**b**) Representation of the interacting residues network between chimeric Bcl-2 (cartoon in green) and venetoclax (sticks in yellow), where the residues Phe112, Thr132 and Glu136 (in red) are specific, when compared to navitoclax binding (sticks in blue). Representation was performed using the PyMOL software (PyMOL Molecular Graphics System, Version 1.8 Schrödinger, LLC).

**Figure 4 ijms-20-00860-f004:**
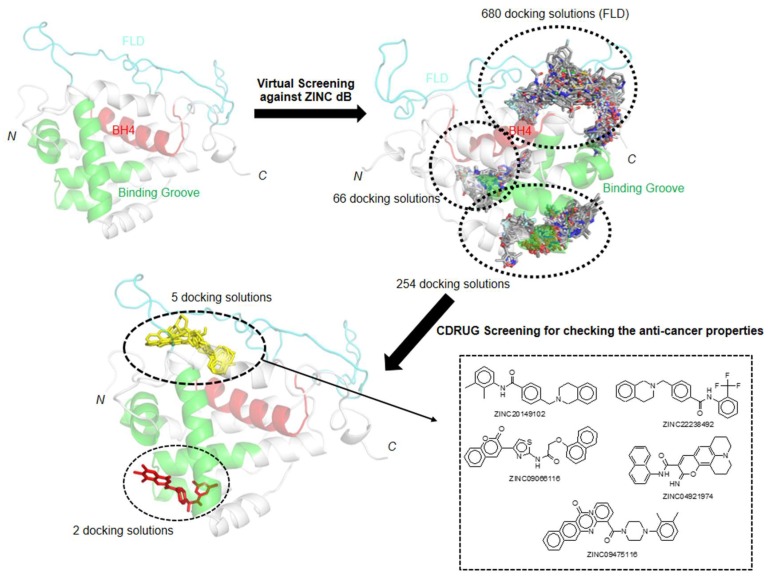
Schematic representation of virtual screening work flow to screen the possible FLD binders for Physiological Bcl-2 form.

**Figure 5 ijms-20-00860-f005:**
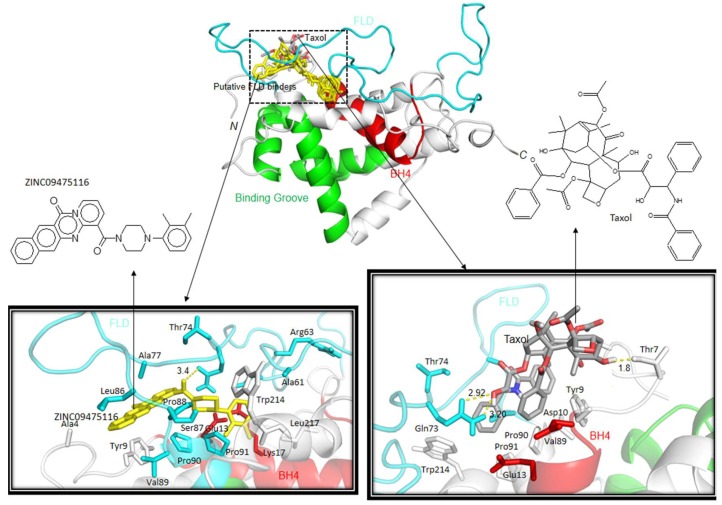
Docking orientations of putative inhibitors and Taxol with the physiological form of Bcl-2. Docking orientations were visualized by PyMOL Program [53]. Chemical scheme of Top1 putative inhibitor ZINC09475116 and Taxol are also presented. (Color codes: Cyan—FLD; Green—Binding groove formed by BH1, BH2 and BH3 domains; Red—BH4 domain).

**Table 1 ijms-20-00860-t001:** List of Bcl-2 nsSNP under study and corresponding binding free energy, potential energy and essential dynamics analysis from the Auto Dock Vina [52] and MD simulations, respectively.

Bcl-2 Wild Type and nsSNP	Venetoclax ΔG (kcal/mol)	Potential Energy (kJ/mol)	ED 2D Projection (nm^2^)
Wild-Type	−9.1	−739988	184.58
G8E	−9.4	−757365	206.09
D34Y	−9.9	−736337	159.06
A43T	−10.2	−757057	156.19
H94P	−9.0	−753942	110.99
L97P	−8.1	−754350	210.22
F104S	−10.5	−757091	199.99
S105F	−10.5	−756523	243.53
S105P	−9.1	−754278	155.57
R129C	−9.1	−757121	167.43
G203S	−9.2	−756615	169.39
R207W	−9.5	−756785	166.20
G233S	−9.3	−757958	177.83

**Table 2 ijms-20-00860-t002:** List of putative and known inhibitors (Taxol) for FLD of the physiological Bcl-2 form. The putative inhibitors were identified from HTVS and docking analysis against the Zinc Database [55] (ranked according to CDRUG [54] P-Value).

Zinc Accession No	Estimated Free Energy of Binding (kcal/mol)	Estimated Inhibition Constant	CDRUG *P* Value
Taxol	−6.7	12.3 μM	0
ZINC20149102	−9.8	66.5 nM	0.0533
ZINC22238492	−9.8	66.5 nM	0.0773
ZINC09066116	−10	46.7 nM	0.0785
ZINC04921974	−10.5	20.1 nM	0.0946
ZINC09475116	−10.9	10.2 nM	0.0976

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
