# Peer review of "Shedding Light on the Interaction of Human Anti-Apoptotic Bcl-2 Protein with Ligands through Biophysical and in Silico Studies"

_ijms, 2019, doi:10.3390/ijms20040860_

Round 1

Reviewer 1 Report

The manuscript by Ramos and collaborators focuses on the interaction of the human anti-apoptotic Bcl-2 protein with ligands using biophysical and in silico approaches. Interactions with venotoclax were characterized with TSA and urea electrophoresis which showed a significant increase in protein stability. Bioinformatics analysis identified deleterious non-synonymous polymorphisms and their impact on ventoclax binding. They also identified new inhibitors from high throughput virtual screening.

The manuscript is well written and conclusions from in silico analysis seem very interesting. This work would clearly benefit from experimental validation (point mutations and/or characterization of putative FLD inhibitors.

Author Response

Response to Reviewer 1 Comments

Point 1: The manuscript by Ramos and collaborators focuses on the interaction of the human anti-apoptotic Bcl-2 protein with ligands using biophysical and in silico approaches. Interactions with venotoclax were characterized with TSA and urea electrophoresis which showed a significant increase in protein stability. Bioinformatics analysis identified deleterious non-synonymous polymorphisms and their impact on ventoclax binding. They also identified new inhibitors from high throughput virtual screening. The manuscript is well written and conclusions from in silico analysis seem very interesting. This work would clearly benefit from experimental validation (point mutations and/or characterization of putative FLD inhibitors.

Response 1:

The work presented here focuses on a well-known inhibitor – venetoclax – and on new putative inhibitors, identified using in silico tools: the experimental data given supports the strong binding interaction between the chimeric form of Bcl-2 (with short flexible loop domain); the in silico data opens a new avenue towards new classes of inhibitors that bind the flexible loop domain. In the future, our goal is to investigate the inhibitory potency of these new putative inhibitors with the physiological form of Bcl-2, not yet available in our lab due to protein stability problems. By that time, we will characterize in more detail the interaction of the ligands with the flexible loop domain, using in vitro assays, site-directed mutagenesis, biophysical and X-ray crystallographic/STD-NMR studies, as suggested by the referee.  

Reviewer 2 Report

In this manuscript, Ramos et al. used biophysical, bioimformatics, molecular modeling approaches to investigate the interaction between Bcl-2 and its specific inhibitor, venetoclax. The authors found that venetoclax binds strongly to Bcl-2 mainly at the BH3 binding groove, and also weakly to the Flexible Loop Domain (FLD), and such binding results in a more stable conformation. It was also found that the conformation of Bcl-2 was not altered upon venetoclax binding, and this binding was not affected by Non-synonymous Single Nucleotide Polymorphisms (nsSNPs). Furthermore, the authors used virtual screening to identify putative Bcl-2 inhibitors that specifically bind to the FLD. Overall, the manuscript presented some novel findings, including the involvement of the FLD in the binding, the effects of nsSNPs, and the putative hits from the virtual screen, that should be useful in drug design against Bcl-2. However, it will strength the conclusions if functional assays are presented.

Major points:

The authors compared the binding of venetoclax to a chimeric Bcl-2, whose FLD contains a stretch of the corresponding region of Bcl-xL, and a “physiological” Bcl-2 (wild-type) in docking and molecular dynamics with computational analysis. However, no significant differences were observed despite the significantly different FLDs. The authors need to explain these results further. It will greatly strengthen the paper if they can test the binding of venetoclax to these two forms of Bcl-2 and examine for a functional difference.  

It is shown that apart from the BH3 groove, venetoclax can also bind to the FLD, although with a much lower affinity. What is the consequence of this binding or the loss of this binding?

As the FLD is known to be flexible, the authors should either directly test the 5 hits that bind the FLD in a protein- or cell-based assay, or explain why the virtual screen is reliable.

Minor points:

TSA should be written in the full name when appearing the first time.

“Chimeric Bcl-2” should be described and explained in the Results section.

The effort to find more candidates that target Bcl-2 is exciting but seems to be very independent of the study on the interaction of venetoclax with Bcl-2. Can the authors give more logical association between the two parts?

For the nsSNPs examined in this study, is any of them reported to affect therapy or prognosis for cancer patients?

The Introduction mentioned Bid as a multi-region protein. That is not correct or at least not accepted by the mainstream. It is a BH3-only protein. Also, the Direct Activation model in which Bim/Noxa recruits Bax/Bak to the mitochondria has been seriously challenged by the O’Neill et al paper (Genes & Development, 2016). The authors should look into and incorporate that line of thinking.

Author Response

Response to Reviewer 2 Comments

Point 1: In this manuscript, Ramos et al. used biophysical, bioimformatics, molecular modeling approaches to investigate the interaction between Bcl-2 and its specific inhibitor, venetoclax. The authors found that venetoclax binds strongly to Bcl-2 mainly at the BH3 binding groove, and also weakly to the Flexible Loop Domain (FLD), and such binding results in a more stable conformation. It was also found that the conformation of Bcl-2 was not altered upon venetoclax binding, and this binding was not affected by Non-synonymous Single Nucleotide Polymorphisms (nsSNPs). Furthermore, the authors used virtual screening to identify putative Bcl-2 inhibitors that specifically bind to the FLD. Overall, the manuscript presented some novel findings, including the involvement of the FLD in the binding, the effects of nsSNPs, and the putative hits from the virtual screen, that should be useful in drug design against Bcl-2. However, it will strength the conclusions if functional assays are presented.

Response 1: The work presented here focus on a well-known inhibitor-venetoclax and on new putative inhibitors, identified using in silico tools. The experimental data supports the strong binding interaction between the chimeric form of Bcl-2 (with a shortened flexible loop domain) reported by Petros et al (Petros, A.M.; Medek, A.; Nettesheim, D.G.; Kim, D.H.; Yoon, H.S.; Swift, K.; Matayoshi, E.D.; Oltersdorf, T.; Fesik, S.W. Solution structure of the antiapoptotic protein bcl-2. Proc. Natl. Acad. Sci. 2001, 98, 3012–3017). The in silico data opens a new avenue towards new classes of inhibitors that bind the flexible loop domain. In the future, our goal is to investigate the inhibitory potency of these new putative inhibitors with the physiological form of Bcl-2, not yet available in our lab due to protein stability problems. By that time, we will characterize in more detail the interaction of the ligands with the flexible loop domain, using in vitro assays, site-directed mutagenesis, biophysical and X-ray crystallographic/STD-NMR studies, as suggested by the referee.  

Point 2: The authors compared the binding of venetoclax to a chimeric Bcl-2, whose FLD contains a stretch of the corresponding region of Bcl-xL, and a “physiological” Bcl-2 (wild-type) in docking and molecular dynamics with computational analysis. However, no significant differences were observed despite the significantly different FLDs. The authors need to explain these results further. It will greatly strengthen the paper if they can test the binding of venetoclax to these two forms of Bcl-2 and examine for a functional difference.

Response 2:

The in silico studies were carried out using the physiological and the chimeric forms of the protein. The physiological form of Bcl-2 has a longer flexible loop, comprising the residues 34-92, which is highly disordered, and interferes with the stability of the protein in vitro. Moreover, it has a transmembrane helix including the residues 212-233. To our knowledge, the full-length (239 amino acid residues) structure of the physiological Bcl-2 form is not yet reported in the PDB, potentially due to this stability issues. This challenge regarding production and manipulation of the physiological Bcl-2 form was circumvented by using the chimeric form, reported by Petros et al (Petros, A.M.; Medek, A.; Nettesheim, D.G.; Kim, D.H.; Yoon, H.S.; Swift, K.; Matayoshi, E.D.; Oltersdorf, T.; Fesik, S.W. Solution structure of the antiapoptotic protein bcl-2. Proc. Natl. Acad. Sci. 2001, 98, 3012–3017). The chimeric form of Bcl-2 has the flexible loop domain replaced by a derivative loop from Bcl-xL, which decreases protein flexibility and enhances stability.

To make this point clear for the reader, we included in the introduction a statement explaining the differences between the chimeric and the rationale behind it (Line No: 115-119).

“In the present work, we explore the human Bcl-2 protein-ligand interactions using biophysical and computational methods: experimental assays were carried out using a chimeric form of Bcl-2 where the flexible loop domain was replaced by a derivative loop from Bcl-xL to overcome protein precipitation, as described by Petros et al[18]; while computational studies were carried out using chimeric and physiological forms, for comparison”.

In our study, in order to compare the chimeric and the physiological forms, we used in silico methods, which allowed investigate the chimeric bcl-2:venetoclax complex and the physiological Bcl-2:venetoclax complex. The molecular docking and molecular dynamics simulation studies suggest that Venetoclax binds effectively to the BH3 binding pocket in both forms of Bcl-2 with significant binding affinity (Physiological Bcl-2 form: -10.24 kcal/mol and chimeric Bcl-2 form: -11.35 kcal/mol) and affects protein stability. We did molecular dynamics simulations for the two forms and, initially the physiological form of Bcl-2 had higher RMSD values than the chimeric form. Nonetheless, after reaching 25 ns, both forms had similar RMSD values. In order to make this point clearer, we changed the text accordingly (line 210-215):

The average RMSD value was from 0.45 to 0.5 nm indicating that the complexes were stable and had converged properly. During the MD simulations we could observe that the physiological form of Bcl-2 showed higher RMSD values than the chimeric form, likely due to the instability conferred by the large flexible loop. Nonetheless, after reaching 25 ns, both systems were equilibrated having similar RMSD values, showing that the binding of the drug molecule is similar in both cases (Figure S3).

In lines 233-236, the same results are explained further:

Essential dynamics analysis supported that the ligand-bound form had slightly higher flexibility for the physiological Bcl-2 compared to the chimeric form due to the presence of a long and flexible loop (Table S5)”.

In lines 181-182

The high number of interaction sites suggests a tight binding between the physiological Bcl-2 and venetoclax.

In fact, it is not surprising that the binding of venetoclax is not affected by the replacement of the FLD domain, Venetoclax binds in the BH3 binding groove, having no significant interaction with the FLD domain. However, in the presence of, for example, p53, it can be expected a decrease in binding affinity, due to the regulatory activity of the FLD, promoted by p53.

Point 3: It is shown that apart from the BH3 groove, venetoclax can also bind to the FLD, although with a much lower affinity. What is the consequence of this binding or the loss of this binding?

Response 3: Venetoclax strongly binds to the BH3 binding groove of Bcl-2 protein with a strong binding affinity. If venetoclax binds to BH3 groove, it can be used as competitive inhibitor. Flexible Loop Domain (FLD) of Bcl-2 is also an essential for regulatory function of cell apoptotic process. If venetoclax binds to FLD, it may be used as non-competitive inhibitor. Our results suggest that binding to BH3 groove is more likely but, both binding sites are possible. In the case of physiological Bcl-2, we reckon that the binding, at the BH3 groove or at the FLD, will inhibit the protein and disable its anti-apoptotic effect.

Point 4: As the FLD is known to be flexible, the authors should either directly test the 5 hits that bind the FLD in a protein- or cell-based assay, or explain why the virtual screen is reliable.

Response 4: The structure-based virtual screening approach is one of the most efficient ways to investigate protein-ligand interactions, displaying overwhelming success in dug design. Several successful molecules were discovered based on this approach such as: Captopril (Angiotensin converting enzyme inhibitor), Dorzolamide (Carbonic anhydrase inhibitor), Nelfinavir (HIV protease inhibitor), Amprenavir (HIV protease inhibitor), Zanamivir (Neuraminidase inhibitor), etc (Kubinyi, H. Success Stories of Computer-Aided Design. In Computer Applications in Pharmaceutical Research and Development; 2006 ISBN 0471737798). It is one of the most widely used techniques to screen and identify putative inhibiting molecules against novel pathogenic proteins. To our knowledge Taxol is the only drug reported which inhibits the function of the FLD (Ferlini et al, Cancer Res. 2009, 69, 6906–6914). Paclitaxel directly binds to Bcl-2 and functionally mimics the activity of Nur77 . Since this plays a crucial role in regulating apoptosis, it should be considered for targeting when searching for Bcl-2 inhibitors. The phosphorylation process of specific residues (T56, T69, S70, T74 and S87) within this region are mainly responsible for the crucial regulatory function (Deng, X.; Gao, F.; Flagg, T.; Anderson, J.; May, W.S. Bcl2’s Flexible Loop Domain egulates p53 Binding and Survival. Mol. Cell. Biol. 2006, 26, 4421–4434). Phosphorylation of S70 essential for the anti-apoptotic function of Bcl-2 protein (Ito, T.; Deng, X.; Carr, B.; May, W.S. Bcl-2 phosphorylation required for anti-apoptosis function. J. Biol. Chem. 1997, 272, 11671–11673). Based on the importance of the FLD, in this study, a high throughput virtual screening (HTVS) identified 5 putative FLD inhibitors from the Zinc database using Auto Dock Vina, showing nanomolar affinity toward the FLD of Bcl-2. Our future work is directed towards validating these inhibitors, either performing in vivo or in vitro assays. At the computational level, we already confirmed the putative anti-cancer activity of these five molecules using the CDRUG web server (Li, G.H.; Huang, J.F. CDRUG: A web server for predicting anticancer activity of chemical compounds. Bioinformatics 2012, 28, 3334–3335). Further experimental studies are planned for the near future.

Point 5: TSA should be written in the full name when appearing the first time.

Response 5: We corrected now and incorporated as Thermal Shift Assay for the first in the manuscript (Line No: 23)

Point 6: “Chimeric Bcl-2” should be described and explained in the Results section.

Response 6: We discussed and explained about Chimeric Bcl-2 in the revised manuscript, at the end of the introduction section (Line No: 115-119):

“In the present work, we explore the human Bcl-2 protein-ligand interactions using biophysical and computational methods: experimental assays were carried out using a chimeric form of Bcl-2 where the flexible loop domain was replaced by a derivative loop from Bcl-xL to overcome protein precipitation, as described by Petros et al[18]; while computational studies were carried out using chimeric and physiological forms, for comparison.”

Point 7: The effort to find more candidates that target Bcl-2 is exciting but seems to be very independent of the study on the interaction of venetoclax with Bcl-2. Can the authors give more logical association between the two parts?

Response 7: The objective of our study is to understand the interaction mechanism of Bcl-2 and ligands using biophysical and in silico studies. Our goal is to characterize a known inhibitor and to find new putative inhibitors.

Venetoclax was designed to bind the BH3 binding groove and act as a competitive inhibitor. This molecule is not deprived of side-effects and drug developers are looking for new molecules that can effectively and selectively bind Bcl-2. As an alternative binding site we suggest the FLD: although not directly involved in Bcl-2:Bax binding, it plays a regulatory role for the interaction, because several phosphorylating residues are found in this region, particularly Ser70. This phosphorylation process is necessary and pivotal for Bcl-2:Bax interaction, according to Ruvolo et al (Ruvolo, P.P.; Deng, X.; May, W.S. Phosphorylation of Bcl2 and regulation of apoptosis. Leukemia 2001, 15, 515–522). For this reason, we consider that searching for Bcl-2 inhibitors that bind to the FLD is of utmost importance, especially when considering combination therapy.

To make this point clearer we changed the text accordingly (line 271-279):

“As mentioned previously, p53 binding to the FLD of Bcl-2 weakens the Bcl2:Bax interaction and results in the enhancement of the apoptotic pathway. The instability, that this region promotes in Bcl-2 is a major limitation for experimental studies on recombinant Bcl-2. Therefore, the chimeric form reported by Petros et al[18], with enhanced stability is used in the present study. In silico studies offer the possibility to investigate the interaction of the FLD with small molecules, paving the way for finding new Bcl-2 inhibitors. These new promising inhibitor candidates can be tested later for in vivo activity and can be considered for combination therapy. Hence, we attempted to find candidates to target the physiological Bcl-2 form, by performing a virtual screening using the Zinc database[32].”

Point 8: For the nsSNPs examined in this study, is any of them reported to affect therapy or prognosis for cancer patients?

Response 8: In this study, we analyzed G8E, D34Y, A43T, H94P, L97P, F104S, S105F, S105P, R129C and G203S nsSNPs. A43T is associated with Esophageal cancer and Hepatocellular carcinoma (HCC) (1. Jain, M; Kumar, S; Lal, P; Tiwari, A; Ghoshal, U.C; Mittal, B. Role of BCL2 ( ala43thr ), CCND1 ( G870A ) and FAS ( A-670G ) polymorphisms in modulating the risk of developing esophageal cancer. Cancer Detect Prev. 2007, 31, 225–232, 2. Abdel-hamid, M.; Shaker, O.G.; Ellakwa, D.E.; Abdel-maksoud, E.F. Detection of BCL2 Polymorphism in Patient with Hepatocellular Carcinoma. Am J Cancer Prev 2015, 3, 27–34. An in vitro study of ala43thr (G128A) polymorphism in Bcl-2 anti-apoptotic gene showed that ala43ala genotype increases the survival of cells, while the threonine variant acts as a suppressed haplotype for the anti-apoptotic factor (Jain, M; Kumar, S; Lal, P; Tiwari, A; Ghoshal, U.C; Mittal, B. Role of BCL2 ( ala43thr ), CCND1 ( G870A ) and FAS ( A-670G ) polymorphisms in modulating the risk of developing esophageal cancer. Cancer Detect Prev. 2007, 31, 225–232).  Abdel-Hamid and his collaborators previously reported that Bcl-2’s A43T polymorphism is associated with the susceptibility to HCC in Egyptian populations and might be used as molecular markers for evaluating HCC risk. To our knowledge, the other nsSNPs have not been study from a functional or structural point of view. Our goal was to evaluate the putative effect of the mutation on protein stability and on venetoclax binding.

Now we incorporated this details in the revised version of manuscript (Line 250-251)

“Among the 12 putative deleterious nsSNPs here predicted, A43 was reported to be associated with Esophageal cancer[42] and Hepatocellular carcinoma (HCC)[43]

Point 9: The Introduction mentioned Bid as a multi-region protein. That is not correct or at least not accepted by the mainstream. It is a BH3-only protein. Also, the Direct Activation model in which Bim/Noxa recruits Bax/Bak to the mitochondria has been seriously challenged by the O’Neill et al paper (Genes & Development, 2016). The authors should look into and incorporate that line of thinking.

Response 9: We agree with the reviewer: Bid is a BH3-only protein and this part in the original version of the manuscript was not properly explained. We changed this paragraph in the revised version as follows (Line 60-79):

B-cell lymphoma 2 (Bcl-2) family proteins are responsible for the homeostasis that dictates cell survival or cell death, through the intrinsic pathway. The first member of this family to be found was the Bcl-2 protein, which was isolated from a t(14;18) chromosomal translocation derived from a human follicular lymphoma by Fukuhara and Rowley[6]. The remaining Bcl-2 family members were added by sequence homology identification, which enabled the postulation of four different domains, named Bcl-2 homology domains (BH1, BH2, BH3, BH4)[7,8]. The Bcl-2 family proteins can be divided into two groups, according to their function: anti-apoptotic and pro-apoptotic. The anti-apoptotic members are Bcl-2, Bcl-w, Bcl-xL, Mcl-1 and A1, which are multi-BH domain proteins, including the four BH domains and a transmembrane domain (TM). While the pro-apoptotic members comprise multi-BH domain (Bax and Bak) and BH3-only proteins (Bad, Bim, Bid, Bik, Bmf, Hrk, Noxa and Puma).

As the cell senses a death signal, the activators BH3-only proteins sequester the anti-apoptotic proteins, such as Bcl-2. Unable to bind the BH3 domain of its pro-apoptotic partners, Bcl-2 is not capable of abolishing the apoptotic pathway. The apoptotic proteins, as Bax, will form homo-oligomers in the mitochondria outer membrane (MOM), promoting its permeabilization to apoptogenic proteins, such as cytochrome c and SMAC[9]. This phenomenon of cytochrome c displacement from the mitochondria to the cytosol leads to the apoptotic events described before. One may consider this mechanism as a population-equilibrium, since the higher availability of pro-apoptotic members causes the progression of apoptosis, while an excess of anti-apoptotic members avoids the apoptotic pathway, through the inhibition of their pro-apoptotic partners”.

Reviewer 3 Report

The result of this paper is the finding that Bcl-2 protein stability is increased upon interaction with the drug venetoclax. The high affinity binding of this drug was shown previously by Souers et al. and the binding site is known. Furthermore, the physiological effect of venetoclax is well studied. The drug is FDA and EMA approved and in use in Europe and USA.

Most of the paper is an in silico investigation into the binding mode of venetoclax which has little consequence as it produces contradictory results and lacks experimental evidence that could have been obtained by mutational and biophysical analysis of the investigated protein.

The here presented data present only an incremental increase in knowledge about the drug venetoclax. Lastly, the experiments lack controls (see point 6 and 8 specific comments).

General comment:

The paper is written for a specific audience familiar with the applied biohpysical and in silico methods. Interested experimentalists with cell biology or biochemistry background will have difficulties following the results and dis. One example is the use of acronyms without introduction, like TSA in the abstract. TSA might be a common technique in crystallography and drug discovery but scientists from other field may not know it. A google search to ‘TSA protein’ yields links to protein bars and the Transportation Security Administration.

Specific comments:

1) Line 55-74: This paragraph is very confusing which is partly due to the terminology. There are lots of different proteins, families, domains, interactions but not well explained. During the second part suddenly other proteins appear that were not introduced. Since it is not the scope of the paper to review all these different protein families, domains and interactions, please reduce this paragraph to what is relevant for this study.

2) Line 63: A transmembrane domain is not a membrane binding domain and vice versa. Proteins with transmembrane domain are integral membrane proteins. Please specify.

3) Line 94-96: Is the MBD or BH3 domain involved in the interaction? Unclear sentences.

4) Line 99: Incorrect use of the word ‘mimetic’. Mimesis means imitation or reproduction. A BH3 mimetic drug would mean ‘a drug that looks like BH3’ but what is meant is a drug that binds to BH3.

5) Line 124: please explain in one sentence what TSA is. The hint that it is a common technique is not helpful.

6) Line 127: All measurements were performed in high concentrations of DMSO that has been shown to have an effect on “protein stability, protein aggregation, and binding of drug compounds” (Tjernberg et al., 2006). As a control, the protein should be measured without DMSO as well.

7) Line 127: Please introduce your chimeric construct and the rationale and consequence of using it instead of the wild type protein. In what respect does it behave differently than the physiological protein?

8) Line 128/Figure 1: What is the control for the addition of venetoclax in high micro-molar concentrations and molar excess? Is the observed stabilization effect specific for this particular protein or also for others? Will it also have the same effect for other BH3 containing proteins?

9) Line 200: “Molecular dynamics presents an exceptional opportunity to study protein dynamics”. Why? Please explain. What is the value of such a statement?

10) Line 203: Please do not use acronyms without introducing them.

11) Line 237: Which 8 in silico programs? What is the basis of the analysis?

12) Line 249-253: The analysis of the 8 in silico programs resulted in prediction that the nsSNPs lead to destabilization of the structure while the MD simulation shows the opposite. Please explain the different types of analysis, how do you explain such a contradiction? If a flexible loop can interfere with the analysis, then how and is it then feasible to do it?

13) Line 265: Please explain why you would like to find new drug candidates. There are already drugs available and studied to great extent.

Author Response

Response to Reviewer 3 Comments

Point 1: The paper is written for a specific audience familiar with the applied biohpysical and in silico methods. Interested experimentalists with cell biology or biochemistry background will have difficulties following the results and dis. One example is the use of acronyms without introduction, like TSA in the abstract. TSA might be a common technique in crystallography and drug discovery but scientists from other field may not know it. A google search to ‘TSA protein’ yields links to protein bars and the Transportation Security Administration.

Response 1:  Thank you very much for considering our manuscript in the International Journal of Molecular Science. The objective of our present study is to understand the interaction mechanism of Bcl-2 and ligand molecules using biophysical and bioinformatics techniques. As pointed by the reviewer, we introduced in the abstract and text the full name of all acronyms.

Point 2: Line 55 to 74: This paragraph is very confusing which is partly due to the terminology. There are lots of different proteins, families, domains, interactions but not well explained. During the second part suddenly, other proteins appear that were not introduced. Since it is not the scope of the paper to review all these different protein families, domains and interactions, please reduce this paragraph to what is relevant for this study.

Response 2: To make this point clearer, we changed the text according to reviewer suggestion (Line 71-79):

As the cell senses a death signal, the activators BH3-only proteins sequester the anti-apoptotic proteins, such as Bcl-2. Unable to bind the BH3 domain of its pro-apoptotic partners, Bcl-2 is not capable of abolishing the apoptotic pathway. The apoptotic proteins, as Bax, will form homo-oligomers in the mitochondria outer membrane (MOM), promoting its permeabilization to apoptogenic proteins, such as cytochrome c and SMAC[9]. This phenomenon of cytochrome c displacement from the mitochondria to the cytosol leads to the apoptotic events described before. One may consider this mechanism as a population-equilibrium, since the higher availability of pro-apoptotic members causes the progression of apoptosis, while an excess of anti-apoptotic members avoids the apoptotic pathway, through the inhibition of their pro-apoptotic partners”.

Point 3: Line 63: A transmembrane domain is not a membrane binding domain and vice versa. Proteins with transmembrane domain are integral membrane proteins. Please specify

Response 3: Based on the previous literatures (1. Cory, S.; Adams, J.M. The bcl2 family: regulators of the cellular life-or-death switch. Nat. Rev. Cancer 2002, 2, 647–656. 2. Reed, J.C.; Zha, H.; Aime-Sempe, C.; Takayama, S.; Wang, H.G. Structure-function analysis of Bcl-2 family proteins. Regulators of programmed cell death. Adv. Exp. Med. Biol. 1996, 406, 99–112. 3. Raghav, P.K.; Verma, Y.K.; Gangenahalli, G.U. Molecular dynamics simulations of the Bcl-2 protein to predict the structure of its unordered flexible loop domain. J. Mol. Model. 2012, 18, 1885–1906. 4. Itchaki, G.; Brown, R.J. The potential of venetoclax (ABT-199) in chronic lymphocytic leukemia. Ther Adv Hematol 2016, 7, 270–287. 5. Cory, S.; Roberts, A.W.; Colman, P.M.; Adams, J.M. Targeting BCL-2-like Proteins to Kill Cancer Cells. TRENDS in CANCER 2016, 2, 443–460. 6. Yip, K.W.; Reed, J.C. Bcl-2 family proteins and cancer. Oncogene 2008, 27, 6398–6406), Bcl-2 family of anti-apoptotic proteins has four BH domains (BH1, BH2, BH3 and BH4) and a carboxy-terminal transmembrane domain.  Now we used uniformly as “Transmembrane domain” in all parts of the manuscript (Line: 66-70)

The anti-apoptotic members are Bcl-2, Bcl-w, Bcl-xL, Mcl-1 and A1, which are multi-BH domain proteins, including the four BH domains and a transmembrane domain (TM). While the pro-apoptotic members comprise multi-BH domain (Bax and Bak) and BH3-only proteins (Bad, Bim, Bid, Bik, Bmf, Hrk, Noxa and Puma).”

Point 4: Line 94-96: Is the MBD or BH3 domain involved in the interaction? Unclear sentences

Response 4: Bcl-2 protein has four Bcl-2 homology domains (BH1, BH2, BH3 and BH4), Flexible Loop domain (FLD) and Transmembrane domain. The BH1-BH3 domains form the hydrophobic groove or binding groove and BH4 stabilizes the structure. Binding groove plays a crucial role for interaction with the pro-apoptotic partners for its anti-apoptotic activity. The transmembrane domain at the C-terminus is not involved in the interaction.

The text was modified accordingly (line 91-97).

“The human Bcl-2 protein has two reported isoforms, alpha and beta, with structures solved by nuclear magnetic resonance (NMR) by Petros et al, in 2001[18]. The most common alpha isoform has 239 amino acids and comprises the four BH regions, a flexible loop domain (FLD) and the TM domain, in a helical bundle fold. In this arrangement, the BH1-BH3 domains form an hydrophobic groove (binding groove or BH3-groove), while BH4 stabilizes the overall structure. The binding groove plays a crucial role in the anti-apoptotic activity of bcl-2 as it accommodates the BH3 domain of the pro-apoptotic partners (Bax/Bak) [19].”

Point 5: Line 99: Incorrect use of the word ‘mimetic’. Mimesis means imitation or reproduction. A BH3 mimetic drug would mean ‘a drug that looks like BH3’ but what is meant is a drug that binds to BH3.

Response 5: This confusion arises from the fact that we hadn’t explained in detail the binding groove of Bcl-2, as mentioned by the reviewer in the previous point. These molecules are called BH3 mimetics since these drugs mimic the BH3 domain of the pro-apoptotic partners.

Based on the previous literatures and studies (1. Itchaki, G.; Brown, R.J. The potential of venetoclax (ABT-199) in chronic lymphocytic leukemia. Ther Adv Hematol 2016, 7, 270–287. 2. Yap, J.L.; Chen, L.; Lanning, M.E.; Fletcher, S. Expanding the Cancer Arsenal with Targeted Therapies: Disarmament of the Antiapoptotic Bcl-2 Proteins by Small Molecules. J. Med. Chem. 2017, 60, 821–838. 3. Chonghaile, T.N.; Letai, A. Mimicking the BH3 domain to kill cancer cells. Oncogene 2008, 27, S149-S157. 4. Vela, L.; Marzo, I. Bcl-2 family of proteins as drug targets for cancer chemotherapy: the long way of BH3 mimetics from bench to bedside. Curr. Opin. Pharmacol. 2015, 23, 74–81. 5. Delbridge, A.R.D.; Strasser, A. The BCL-2 protein family, BH3-mimetics and cancer therapy. Cell Death Differ. 2015, 22, 1071–1080. 6. Billard, C. BH3 Mimetics: Status of the Field and New Developments. Mol. Cancer Ther. 2013, 12, 1691–1700), we used the same terminology.

The BH3 mimetics concept was responsible for the development of small molecules capable of mimicking the function of BH3-only proteins and thus inducing the cell apoptotic process. Venetoclax is a BH3 mimetic approved drug for the treatment of Chronic lymphocytic leukemia (1. Itchaki, G.; Brown, R.J. The potential of venetoclax (ABT-199) in chronic lymphocytic leukemia. Ther Adv Hematol 2016, 7, 270–287, 2. S. Soderquist, R.; Eastman, A. BCL2 Inhibitors as Anticancer Drugs: A Plethora of Misleading BH3 Mimetics. Mol. Cancer Ther. 2016, 15,2011-2017).

In the revised manuscript, the details on the BH3 groove and binding partners is clearer and understandable for the readers.

Point 6: Line 124: please explain in one sentence what TSA is. The hint that it is a common technique is not helpful.

Response 6: TSA stands for Thermal Shift Assay, now we explained clearly, it is one of the commonly used biophysical techniques to understand the interaction between protein and ligand by measuring the melting temperature (Tm) value of the protein and of the protein-ligand complex. Moreover, this biophysical screening can be used for the discovery of small-molecule ligands against various protein in the area of Drug designing and discovery process. Now we explained more clearly in the revised version of manuscript (Line: 128-130).

Thermal shift assay (TSA) is a fluorescence-based technique, routinely used in protein purification and crystallization, to assess protein stability in different conditions, namely in the presence of ligands, by following the protein thermal denaturation process”

Point 7: Line 127: All measurements were performed in high concentrations of DMSO that has been shown to have an effect on “protein stability, protein aggregation, and binding of drug compounds” (Tjernberg et al., 2006). As a control, the protein should be measured without DMSO as well.

Response 7: A control was performed without DMSO, showing virtually the same melting temperature as the chimeric Bcl-2 protein in 2% DMSO - (included now in the main text, Line:132-133 and Figure 1)

Line: 147-152

Figure 1. (a) TSA of chimeric Bcl-2 and incubated with venetoclax in different protein-ligand molar ratios: Bcl-2 in red, Bcl-2 with 2% DMSO in yellow, 1:2 in green, 1:5 in blue, 1:10 in pink color. Upon ligand incubation, the Bcl-2 Tm shifted significantly toward higher temperatures, representing a DTm = 18° C. (b) Urea polyacrylamide gel of (1) Ligand free form of chimeric Bcl-2 and (2) incubated with chimeric Bcl-2:venetoclax. A great difference in electrophoretic mobility is observed in the presence of ligand, indicating relevant modifications in protein folding

Point 8: Line 127: Please introduce your chimeric construct and the rationale and consequence of using it instead of the wild type protein. In what respect does it behave differently than the physiological protein?

Response 8:

The in silico studies were carried out using the physiological and the chimeric forms of the protein. The physiological form of Bcl-2 has a longer flexible loop, comprising the residues 34-92, which is highly disordered, and interferes with the stability of the protein in vitro. Moreover, it has a transmembrane helix including the residues 212-233. To our knowledge, the full-length (239 amino acid residues) structure of the physiological Bcl-2 form is not yet reported in the PDB, potentially due to this stability issues. This challenge regarding production and manipulation of the physiological Bcl-2 form was circumvented by using the chimeric form, reported by Petros et al (Petros, A.M.; Medek, A.; Nettesheim, D.G.; Kim, D.H.; Yoon, H.S.; Swift, K.; Matayoshi, E.D.; Oltersdorf, T.; Fesik, S.W. Solution structure of the antiapoptotic protein bcl-2. Proc. Natl. Acad. Sci. 2001, 98, 3012–3017). The chimeric form of Bcl-2 has the flexible loop domain replaced by a derivative loop from Bcl-xL, which decreases protein flexibility and enhances stability.

To make this point clear for the reader, we included in the introduction section a short sentence explaining the differences between the chimeric and the rationale (line: 115-119).

“In the present work, we explore the human Bcl-2 protein-ligand interactions using biophysical and computational methods: experimental assays were carried out using a chimeric form of Bcl-2 where the flexible loop domain was replaced by a derivative loop from Bcl-xL to overcome protein precipitation, as described by Petros et al[18]; while computational studies were carried out using chimeric and physiological forms, for comparison.”

Point 9: Line 128/Figure 1: What is the control for the addition of venetoclax in high micro-molar concentrations and molar excess? Is the observed stabilization effect specific for this particular protein or also for others? Will it also have the same effect for other BH3 containing proteins?

Response 9: For the TSA experiments we used DMSO as control. We consider that the fluorescence signal wouldn’t have the same profile as a protein denaturing curve: only a uniform variation in intensity across all temperatures should be observed. Unlike the protein thermal denaturation process, venetoclax wouldn’t display a temperature-dependent behavior of exposing hydrophobic moieties to the fluorophore – which is the process that in the presence of proteins leads to the fluorescence signal in the TSA. The effect of venetoclax on the protein stability is clear for Bcl-2 and should be similar to the other proteins from this family that also have the hydrophobic binding groove. This is one reason why specificity is difficult to achieve for BH3 mimetics as venetoclax and other similar ligands.

Point 10: Line 200: “Molecular dynamics presents an exceptional opportunity to study protein dynamics”. Why? Please explain. What is the value of such a statement?

Response 10: We have re-written this sentence in the revised manuscript explaining why this method is important and what information can be retrieved (Lines 200-208)

“Molecular dynamics presents an opportunity to study protein dynamics in a time- and cost-efficient way, adding to the knowledge retrieved from docking calculations, where the target is usually considered as a rigid identity. In this way, MD simulation is an interesting computational tool to understand the stability, flexibility, folding, and the dynamic behavior of proteins and various bio-molecular complexes (protein-protein, protein-small molecule, protein-carbohydrate and protein-lipid) at the same time with regular time scale. Several essential structural parameters are obtained from this analysis such as root mean square deviation (RMSD), root mean square fluctuation (RMSF), radius of gyration (Rg), intra-molecular hydrogen bonds, inter-molar hydrogen bonds, solvent accessible surface area (SASA), secondary structure and residue occupancy probability.”

Point 11: Line 203: Please do not use acronyms without introducing them.

Response 11: We have written properly in the revised version of manuscript.

Point 12: Line 237: Which 8 in silico programs? What is the basis of the analysis?

Response 12: The programs are listed in the “materials and methods” and properly referenced: I-Mutant, Panther, SNP&Go, SIFT, Provean, Polyphen 2.0, nsSNPAnalyzer and PhD-SNP. The basis of analysis is to predict the putative phenotypic effects (deleterious or non-deleterious) of nsSNPs of Bcl-2 using the afore-mentioned programs.  These programs are mainly based on machine learning methods.  We classified the nsSNPs in to deleterious or non-deleterious by comparing results obtained from all the programs and when concordance was obtained for at least 5 out of 8 programs used.

This information was included in the main text in lines 246-248:

Concordance analysis using 8 different in silico programs (listed in the Materials and Methods section) was used to understand the functional consequences and putative phenotypic effects of nsSNPs, which were classified as deleterious or non-deleterious.”

And in line 499-508:

After gathering 39 different Bcl-2 nsSNPs, a concordance analysis was performed using 8 different programs (I-Mutant[76], Panther[77], SNP&Go[78], SIFT[79], Provean[80], Polyphen 2.0[81], nsSNPAnalyzer[82] and PhD-SNP[83]), in order to assess the potential phenotypic effect (deleterious or non-deleterious) of these mutations. These programs attempt to predict the SNPs impact on protein stability and function, using the ProTherm[84] database (Thermodynamic database for Proteins and Mutants, URL: http://www.abren.net/protherm) and two computational methods, namely machine learning and hidden Markov models (HMM). We classified the nsSNPs in to deleterious or non-deleterious by comparing results obtained from all the programs and when concordance was obtained for at least 5 out of 8 programs used”. 

Point 13: Line 249-253: The analysis of the 8 in silico programs resulted in prediction that the nsSNPs lead to destabilization of the structure while the MD simulation shows the opposite. Please explain the different types of analysis, how do you explain such a contradiction? If a flexible loop can interfere with the analysis, then how and is it then feasible to do it?

Response 13: The 8 programs predict the functional consequences and putative phenotypic effects of nsSNPs, classifying the variants as deleterious and non-deleterious. After predicting which ones would have a deleterious effect, we analyzed the gain or loss of post-translation modifications, impact on overall structure, and on venetoclax binding. The results suggest that some of the nsSNPs gain flexibility when compared with native Bcl-2. These results were not clearly explained in the text, so we changed the manuscript accordingly (Lines: 259-266)

“The MD simulation analysis results (Table S3) suggest that G8E, L97P, F104S, S105F structures are generally more flexible than the physiological form of Bcl-2, with variable interdomain distances. Moreover, the impact of the nsSNPs on venetoclax binding was also investigated, since a specific phenotype may interfere with the effectiveness of a standard treatment, especially since some of the deleterious nsSNPs here suggested are involved in Bcl-2:venetoclax binding (F104 and G203). The binding free energies predicted using Auto Dock Vina[51] docking calculations (Table 1) showed that venetoclax binding is generally favored regardless of the presence of the mutations, probably due to the large number of interaction sites between Bcl-2 and this drug.”

Point 14: Line 265: Please explain why you would like to find new drug candidates. There are already drugs available and studied to great extent.”

Response 14: Venetoclax is an extensively used drug, that got FDA approval in 2016. It was derived from the parent compound navitoclax, based on Bcl-2:navitoclax interaction studies. Even though venetoclax is effective, it is not deprived of side effects and medicinal chemists are continually looking for new drug candidates.

The interaction of ligands with the FLD is very much unexplored, probably due to the hurdle of the instability of the physiological form, which leads to the study of only the chimeric form, without the FLD. Apart from BH domains, FLD domain is also essential to regulate the Bcl-2:Bax interactions. To our knowledge in the present study first time we utilized virtual screening approaches to identify the putative FLD binders which exactly oriented as taxol (only reported FLD inhibitor so far) (Ferlini, C.; Cicchillitti, L.; Raspaglio, G.; Bartollino, S.; Cimitan, S.; Bertucci, C.; Mozzetti, S.; Gallo, D.; Persico, M.; Fattorusso, C.; et al. Paclitaxel directly binds to Bcl-2 and functionally mimics activity of Nur77. Cancer Res. 2009, 69, 6906–6914). Our future work is directed to evaluate anti-cancer properties of FLD binders through experimental studies. To make this point clear, we changed the text accordingly (line 271-288)

“2.4. Virtual screening for physiological Bcl-2: targeting the FLD

As mentioned previously, p53 binding to the FLD of Bcl-2 weakens the Bcl2:Bax interaction and results in the enhancement of the apoptotic pathway. The instability that this region promotes in Bcl-2 is a major limitation for experimental studies on recombinant Bcl-2. Therefore, the chimeric form reported by Petros et al[18], with enhanced stability is used in the present study. In silico studies offer the possibility to investigate the interaction of the FLD with small molecules, paving the way for finding new Bcl-2 inhibitors. These new promising inhibitor candidates can be tested later for in vivo activity and can be considered for combination therapy. Hence, we attempted to find candidates to target the physiological Bcl-2 form, by performing a virtual screening using the Zinc database[32]. We visualized 1000 possible docking solutions of various molecules in PyMOL[52] to locate the binding modes in the physiological Bcl-2 structure. Among the docking solutions, 68% of molecules were oriented towards the FLD, whereas the remaining 32% were located in the binding groove (formed by BH1, BH2 and BH3 domains) of Bcl-2. The docking solutions were submitted to the CDRUG webserver[53] to predict their putative anti-cancer properties. Among the 7 molecules that might have possible anti-cancer properties, 5 have a higher affinity for the FLD and only 2 are oriented in the binding groove. The 5 hits that bind to the FLD are in the same orientation as taxol, known to bind to the domain[31], suggesting that these molecules will also impair the FLD function (Figures 4 and 5)”.

The methodology part of virtual screening was not included in the submitted version, now we incorporated (line: 460-465):

“The Drug discovery@TACC web portal (https://drugdiscovery.tacc.utexas.edu/#/home) was used to perform the high throughput virtual screening analysis of physiological Bcl-2 form against the Zinc database [32] (library of ~642,759 drug-like molecules) using Auto Dock Vina[51] to identify the putative FLD inhibitors. The steps involved for protein preparation was mentioned above. Once virtual screening results were obtained, they were analyzed by using CDRUG webserver [53] and PyMOL[52] program”.

Round 2

Reviewer 2 Report

The authors made good improvements in their writing, making it much easier to read. However, they did not correctly introduce the actions of the Bcl-2 family proteins, especially the BH3-only proteins, as described in the literature. The current consensus is that the BH3-only proteins are classified into the “activators and sensitizers”, with the activators being able to directly activate Bax or Bak, the effector proteins, and the sensitizers being able to only sequester the anti-apoptotic Bcl-2 proteins. The O’Neill et al. paper is noteworthy because it seriously challenged this consensus. The authors failed to properly describe the perceived actions of the different BH3-only proteins, which are the origin of the BH3 mimetics used in this manuscript.

Author Response

Response to Reviewer 2 Comments

Point 1: The authors made good improvements in their writing, making it much easier to read. However, they did not correctly introduce the actions of the Bcl-2 family proteins, especially the BH3-only proteins, as described in the literature. The current consensus is that the BH3-only proteins are classified into the “activators and sensitizers”, with the activators being able to directly activate Bax or Bak, the effector proteins, and the sensitizers being able to only sequester the anti-apoptotic Bcl-2 proteins. The O’Neill et al. paper is noteworthy because it seriously challenged this consensus. The authors failed to properly describe the perceived actions of the different BH3-only proteins, which are the origin of the BH3 mimetics used in this manuscript.

Response 1: Based on the reviewer suggestions, we revised this part in the manuscript (Line: 68-76).

“The pro-apoptotic members comprise multi-BH domain effector proteins (Bax and Bak) and BH3-only proteins (Bad, Bim, Bid, Bik, Bmf, Hrk, Noxa and Puma) [9,10]. As the cell senses a death signal, BH3-only proteins, called sensitizers, sequester the anti-apoptotic proteins, such as Bcl-2. Unable to bind the BH3 domain of its pro-apoptotic partners, Bcl-2 is not capable of abolishing the apoptotic pathway. The pro-apoptotic proteins Bax/Bak form homo-oligomers in the mitochondria outer membrane (MOM), promoting its permeabilization to apoptogenic proteins, such as cytochrome c and second mitochondria-derived activator of caspases (SMAC)[9]

Reviewer 3 Report

The authors addressed all concerns appropriately. The manuscript is much improved, makes more sense and is understandable for a broader audience; reads well.

Author Response

Response to Reviewer 3 Comments

Point 1: The authors addressed all concerns appropriately. The manuscript is much improved, makes more sense and is understandable for a broader audience; reads well.

Response 1: The authors appreciate the reviewer’s contribution to the improvement of the manuscript in terms of its coherence, transparency and scientific significance.